# Social exclusion and the perspectives of health care providers on migrants in Gauteng public health facilities, South Africa

Janine A. White[1]*, Duane Blaauw[2], Laetitia C. Rispel[3]

**1** School of Public Health, Faculty of Health Sciences, University of the Witwatersrand, Johannesburg, South Africa, **2** Centre for Health Policy, School of Public Health, Faculty of Health Sciences, University of the Witwatersrand, Johannesburg, South Africa, **3** Centre for Health Policy & South African Research Chairs Initiative, School of Public Health, Faculty of Health Sciences, University of the Witwatersrand, Johannesburg, South Africa

* Janine.white@wits.ac.za

## Abstract

### Background

Universal health coverage (UHC) for all people, regardless of citizenship, is a global priority. Health care providers are central to the achievement of UHC, and their attitudes and behaviour could either advance or impede UHC for migrants. Using a social exclusion conceptual framework, this study examined the perspectives of health care providers on delivering health services to migrants in public health facilities in Gauteng Province, South Africa.

### Methods

We used stratified, random sampling to select 13 public health facilities. All health care providers working in ambulatory care were invited to complete a self-administered questionnaire. In addition to socio-demographic information, the questionnaire asked health care providers if they had witnessed discrimination against migrants at work, and measured their perspectives on social exclusionary views and practices. Multiple regression analysis was used to identify predictors of more exclusionary perspectives for each item.

### Results

277 of 308 health care providers participated in the study–a response rate of 90%. The participants were predominantly female (77.6%) and nurses (51.9%), and had worked for an average of 6.8 years in their facilities. 19.2% of health care providers reported that they had witnessed discrimination against migrants, while 20.0% reported differential treatment of migrant patients. Exclusionary perspectives varied across the different items, and for different provider groups. Enrolled nurses and nursing assistants were significantly more exclusionary on a number of items, while the opposite was found for providers born outside South Africa. For some questions, female providers held more exclusionary perspectives and this was also the case for providers from higher levels of care.

**Data Availability Statement:** The Human Research Ethics Committee (HREC) (Medical) of the University of the Witwatersrand in Johannesburg has imposed restrictions on the data, because it

contains sensitive and confidential information on health care providers. Researchers who meet the criteria for access to confidential and sensitive data can contact the university's senior information scientist (nina.lewin@wits.ac.za), or the HREC administrator (zanele.ndlovu@wits.ac.za).

**Funding:** This research was funded by the Atlantic Philanthropies (https://www.atlanticphilanthropies. org/), grant no. 21408, and Professor Rispel's Research Chair funded through the National Research Foundation of South Africa (https://www. nrf.ac.za/). The views expressed in this study are those of the authors. The funders had no role in study design, data collection and analysis, decision to publish, or preparation of the manuscript.

**Competing interests:** The authors have declared that no competing interests exist.

## Conclusion

Health care providers are critical to inclusive UHC. Social exclusionary views or practices must be addressed through enabling health policies; training in culture-sensitivity, ethics and human rights; and advocacy to ensure that health care providers uphold their professional obligations to all patients.

## Introduction

This millennium has been marked by mass migration [1]. In 2019, an estimated 70 million people were displaced globally [2]. In this paper, migrants refer to people who have moved across an international border away from their habitual place of residence, regardless of their legal status, causes of the movement, or whether it was voluntary or involuntary [3]. Worldwide, the unmet health needs of migrants and their lack of access to essential health services are of concern [4]. Consequently, a major global priority is to achieve universal health coverage (UHC) for all people, regardless of citizenship [4]. UHC implies the mainstreaming of migrant health into country-level reform agendas, the promotion of migrant-sensitive health policies [4], and the development of responsive health systems [5]. Human resources for health (HRH) are central to the achievement of UHC, because health care providers are the personification of responsive health systems [6]. Hence, their attitudes, behaviour or practices could either advance or constrain the achievement of UHC for vulnerable individuals, such as migrants [7].

In South Africa there is contestation about the number of migrants, but the 2011 census estimated around 2.2 million immigrants, with a total population of 51.8 million people [8]. In 2020, the International Organization of Migration (IOM) estimated this number to be around 4 million; and highlighted that South Africa was one of the top 20 destinations for migrants, due to an increase in intraregional migration [9]. According to the IOM, the majority of migrants in South Africa came from neighbouring Mozambique and Zimbabwe [9].

Legally, there is a constitutional right to health care for all individuals regardless of nationality, but access to health services for migrants is complex, especially for those without formal documentation [7]. This is partly due to the significant challenges faced by government in providing high-quality health care in the public health sector [10]. At the health facility level, the decisions of hospital managers or administrators could exclude migrants from health care, which is contrary to the Constitution and a violation of their human rights [11]. The proposed National Health Insurance (NHI) system is the country's primary UHC reform, aimed at addressing the entrenched inequities in its two-tiered health system [12]. Although the NHI policy document lacks clarity on health care for migrants and refugees [12], Chapter 2 of the 2019 NHI bill makes provision for complete cover for permanent residents, refugees, and asylum seekers, but only limited cover for "illegal foreigners" [13][p.8].

Research on migration has focused on the legal instruments for protecting the human rights of migrants and refugees [14], health inequities and unmet health needs of migrants and refugees [15], health policy or system deficiencies [5], and migrant or refugee experiences and perceptions of health services in the host countries [16]. A systematic review on health care provider experiences of providing care to migrants and refugees found major challenges related to diverse cultural beliefs, limited institutional capacity, and the inconsistency between health professional ethics and country-specific legislation that often limits migrants' right to health care [17]. A 2017 systematic review of health professionals in primary health care (PHC) settings providing care to refugees and asylum seekers found that political decisions

affect frontline clinical practice, resourcing priorities, health professional roles and healthcare access [18]. The health professionals reported that encounters with refugees and asylum seekers were influenced by cultural differences, and a lack of knowledge of the more complex health problems. They were also exacerbated by health system challenges such as a lack of training, insufficient time or professional support to manage complex health problems, referral difficulties, increased costs, and staff shortages [18]. All of these health system challenges were experienced within a fluid and changing policy environment, and widespread hostility of policymakers to migrants [18, 19].

In Africa, a 2018 WHO report underscored the dearth of empirical information on health care to migrants and the ethical responsibilities or professional duties of health care providers [20]. In South Africa, Matlin *et al* [7] have pointed out that despite an enabling legal framework, health care access for migrants is variable in practice and influenced by health system factors, health managers' responsiveness and xenophobic attitudes by health professionals. In a 2011 qualitative study with Zimbabwean migrants in Cape Town and Johannesburg, Crush and Tawodzera [21] coined the term "medical xenophobia", defined as the "negative attitudes and practices of health sector professionals and employees towards migrants and refugees on the job" [p.655]. Medical xenophobia included the insistence by managers or health care providers that patients show identity documentation prior to receiving care [21]. It also included delay or denial of treatment on the basis of nationality, refusal to communicate with patients in a common language (such as English) or to allow the use of translators, and/or verbal abuse and xenophobic statements and insults [21]. A 2017 qualitative study in Durban, South Africa described the medical xenophobia faced by refugees from the Democratic Republic of Congo (DRC) including the insistence on documentation, insensitive comments and other discriminatory practices from providers [22]. Another small qualitative study that explored the experiences of eight women refugees and their attempts at utilising reproductive health care services in Durban's public sector also reported incidents of medical xenophobia [23]. A final small ethnographic study with 21 HIV positive Mozambicans found that their access to health care services was constrained due to structural vulnerabilities and HIV stigma [24].

However, all of these South African studies were qualitative in design and none of them focused on health care providers. The aim of this study was to examine the perspectives of health care providers on delivering health services to migrants in public health facilities in the Gauteng Province of South Africa. A quantitative approach allowed us to obtain a representative view on health care providers' perspectives on social exclusionary views or practices; and to investigate the relationships between demographic; the type of facility and health care providers' perspectives or practices. The paper contributes to an emerging literature that examines quality UHC for migrants from the perspective of health care providers.

## Material and methods

### Conceptual framework

In this paper, we draw on the social exclusion conceptual framework of the Social Exclusion Knowledge Network (SEKN) to examine the experiences and perspectives of health care providers on migrants utilising public health services in the Gauteng Province of South Africa [25]. The concept and measurement of social exclusion remains contested, illustrated by a 2019 scoping review that highlights different definitions and variations in the measurement of social exclusion [26].

The SEKN defines social exclusion as the "dynamic, multidimensional processes driven by unequal power relationships, interacting across economic, political, social and cultural dimensions and at individual, household, group, community, country and global levels" [25][p36]. In

this study, we examined social exclusion within the health system, specifically the perspectives of health care providers (the personification of the health system) and migrants (a potentially excluded group).

A relational approach to social exclusion has a number of advantages [25]. Firstly, it underscores the complexity and dynamics of the notion of social exclusion embedded in relationships. Secondly, it highlights the salience of identity (in this case nationality) that serves to exclude migrant patients. Thirdly, a relational approach enables an exploration of the linkages between social exclusion and human rights (e.g. non-discrimination). Lastly, a relational approach can elucidate the perspectives of health care providers of migrant patients that generate and/or sustain broader exclusionary processes [25].

## Study design and setting

This is a cross-sectional analytical study conducted in public health care facilities in the Gauteng Province of South Africa.

The study setting was all the public health care facilities in Gauteng Province. The province is the most densely populated in South Africa, with an estimated total population of 14.7 million [27]. In 2018, Gauteng Province was host to the largest proportion of migrants (47.5%) in South Africa [27].

In Gauteng Province, the public health care system consists of four central hospitals, that provide highly specialised quaternary and/or tertiary services, serve as referral hospitals for lower level facilities, and are attached to university health science faculties that train health professionals [28]. There are also two regional tertiary hospitals that are attached to health science faculties and provide some tertiary and other specialised services, and nine regional hospitals that provide specialised secondary services in internal medicine, general surgery, paediatrics, obstetrics and gynaecology and general surgery. The province has one specialised mother-and child-hospital that functions at the level of a regional hospital, with some tertiary services. The 11 district hospitals in the province provide general, inpatient hospital services, and the six specialised hospitals provide psychiatric services, tuberculosis services, infectious diseases and rehabilitation services [28]. The primary health care (PHC) system consists of a network of 30 community health centres (CHCs) and 290 PHC clinics that provide ambulatory care services. The CHCs are open 24 hours per day, seven days per week, while the PHC clinics are open during office hours from Monday to Friday.

The study population consisted of all health care providers that provide ambulatory care services in Gauteng public health facilities. This included medical doctors (both generalists and specialists), professional nurses (with four years of training), enrolled nurses (with two years of training), and nursing auxiliaries or assistants (with one year of training), dentists, occupational therapists, physiotherapists, and pharmacists [29]. We obtained data on health professionals from the Gauteng Department of Health, which showed that in 2018 there were a total of 5102 medical doctors, 12058 professional nurses, 6424 enrolled nurses, 6050 nursing assistants, and 3288 allied health professionals.

## Sampling of facilities

We used stratified, random sampling to select the public health care facilities from the master list of health care facilities in Gauteng Province (obtained from the Gauteng Department of Health). Facilities were stratified by type as follows: central hospital, regional tertiary hospital, regional hospital, district hospital, community health centres, PHC clinics, and a mother and child hospital. We selected two facilities randomly from each stratum, except in the case of the

mother and child hospital, where there is only one. Hence, we sampled 13 public health care facilities in Gauteng.

## Measures

We designed a self-administered questionnaire that obtained information on the socio-demographic profile of health care providers, and that measured social exclusionary views or practices. The questions on social exclusion were based on an extensive literature review and drew on the SEKN conceptual framework [25].

The socio-demographic questions elicited information on age, gender, marital status, category of health care provider, and number of years worked in the health care facility. We measured social exclusionary practices and views among health care providers in two ways. Firstly, health care providers were asked if they had witnessed discrimination or differential treatment of migrants at work (two separate questions measured by yes or no response). Secondly, the health care providers were asked to rate their agreement with seven statements on social exclusionary views or practices (seven questions measured on a seven-point Likert scale from strongly disagree (1) to strongly agree (7)). The seven statements intended to measure the social exclusionary views or practices of health care providers, focused on examples of medical xenophobia, health professional obligations in relation to migrants, and coverage of migrants in the NHI system. Three of the questions were phrased positively (where higher agreement indicates less exclusionary attitudes), and four were phrased negatively (lower agreement indicates less exclusionary attitudes).

We piloted the questionnaire with five health care providers of different categories at a hospital, clinic and community health centre that were not part of the selected facilities to determine clarity of questions and the time taken to complete the questionnaire. Based on the feedback we received, no changes to the provider SAQ were required.

## Preparation for data collection

Prior to data collection, we obtained permission for the study from the relevant authorities (Gauteng Department of Health (GDoH), and the City of Johannesburg), and from each of the sampled health facilities. The ethical approval from the Human Research Ethics Committee (HREC) (Medical) was also included.

The principal researcher (JW) included a cover letter or email to the documents. This cover letter highlighted the following: that 3 days would be selected randomly for data collection, that health care providers on duty in ambulatory care would be approached, and invited to participate in the study, that participation was voluntary, and that the principal researcher and the research team will not interfere with patient care or the duties of health workers.

Due to the number of health care facilities in the study, the principal researcher recruited fieldworkers to assist with data collection. The principal researcher ensured prior training of each of the other members of the research team, using a detailed field manual that stressed ethical conduct, professionalism, and confidentiality. The research team consisted of three South Africans, one national from Ghana and two nationals from Zimbabwe.

The principal researcher liaised with the health facility manager at each hospital, or clinic prior to data collection. Once the data collection days were selected randomly for each facility, the principal researcher ensured that the manager in charge of the facility was aware of the selected days, and that the research team would arrive before 7 am in the morning. On the survey days, the principal researcher, announced the presence of the research team and introduced the team to the manager on duty.

## Data collection

We conducted the study between April and December 2018 at the 13 selected public health care facilities. For primary health care clinics, we selected three separate days randomly between Monday and Friday. In the case of community health centres and hospitals, we selected two days randomly between Monday and Friday, and one day randomly on the weekend.

The research team recruited health care providers on the randomly selected fieldwork days at each of the selected facilities. The eligibility criteria for participation in the study was working in ambulatory care in the facility, which meant outpatient or emergency medical department for the hospitals. A member of the research team approached an eligible health care provider during their tea or lunch break, explained the study verbally, and offered each participant the study information sheet. The information sheet contained background on the study, the confidential and voluntary nature of the study, the participant's rights, including withdrawal at any point, no incentives or penalties, and the contact details of the HREC and the study supervisor. All participants gave voluntary, informed, written consent.

We used mobile device/tablet for data collection, with the self-administered questionnaire preloaded on the tablet. Once the initial approach was made to the eligible health care providers and they agreed to participate in the study, the tablet was handed to the health care provider to complete the survey, with direct data entry into Research Electronic Data Capture (REDCap). The latter is a secure web-based programme hosted at the University of Witwatersrand [30]. Each participant had to provide written consent, by indicating yes or no on the tablet. A no response would take them out of the survey. During completion of the survey, a member of the research team stood a distance away from participants to allow for privacy, but close enough to answer any questions. No health facility supervisor or manager was present during survey completion. The health care providers did not receive anything for participating in the study.

## Statistical analysis

We used Stata® 15 to analyze the data. Frequency tabulations were done to describe the socio-demographic and employment characteristics of the study participants.

The analysis took account of the complex sampling design using the svyset command in Stata. We used the GDoH data on all health professionals by category and health facility to calculate the weights for analysis. All analyses were weighted to reflect the distribution of health care providers, by type of health facility and health worker category, at the provincial level. For analysis we combined: central hospitals, regional tertiary hospitals and specialised mother and child facility into one category, called "tertiary hospitals"; and clinics and community health centres into "primary health care (PHC) facilities". In the case of health care provider, we combined all the enrolled nurses and nursing assistants into one category, called enrolled nurses and nursing assistants. We also adjusted standard errors for clustering at the health facility level.

We used frequency tabulations to show the proportion of health care providers reporting that they witnessed discrimination against, or differential treatment of, migrants. We computed the mean and standard deviations for the 7-point Likert scale items that measured social exclusionary views or practices. Bivariate analysis was done to investigate the relationship between the socio-demographic and employment characteristics of health care providers and each of the social exclusionary items. All the factors found to be statistically significant at a conservative level of 20% level were included in the multiple regression models, which were evaluated for each of the items separately. All tests were conducted at a 5% significance level.

### Ethical considerations

We obtained ethical approval from the HREC (Medical) of the University of the Witwatersrand in Johannesburg (Certificate #: M170988). We also obtained permission from the Gauteng Department of Health through the National Health Research Database (NHRD reference #: GP_201804_019). All participants received a detailed study information sheet, and provided written consent, via REDCap (Research Electronic Data Capture) [30]. We complied with the Singapore Declaration of research integrity [31] and adhered to all ethical procedures, including informed consent, voluntary participation, confidentiality and anonymity.

## Results

We obtained a 89.9% response rate, with 277/308 health care providers participating in the study (Table 1). The study participants were predominantly women (77.6%), with a mean age of 36.2 (SD 11.4), and a median age of 33 years (range 19–68). The mean age of professional nurses was 45.0 years (SD 11.9), while allied health professionals had a mean age of 28.8 (SD 6.9). Nurses constituted the largest group of study participants (51.9%) and the overwhelming majority were South African (94.8%). A quarter of all study participants worked at central hospitals (25.1%) (Table 1). The mean years worked at any of the selected facilities was 6.8 years (SD 8.4) (Table 1).

### Health care provider reported discrimination or differential treatment

Of the health care providers surveyed, 19.2% reported that they had witnessed discrimination and 20.0% reported that they witnessed differential treatment of migrants in their work settings (Fig 1). Medical doctors reported witnessing discrimination (31.4%) or differential treatment (31.4%) more frequently than other categories of health professionals (Fig 1). The difference between medical doctors and the other professional groups combined was statistically significant for witnessing discrimination ($x^2$ = 9.89, p = 0.005), but not statistically significant for witnessing differential treatment ($x^2$ = 4.35, p = 0.05).

### Social exclusionary views or practices

Table 2 shows health care providers' mean scores for social exclusionary views or practices, for all providers combined and by socio-demographic and employment characteristics. The items are scored from 1 (strongly disagree) to 7 (strongly agree) so the mean scores should be interpreted in comparison to a score of 4, the midpoint of the scale. Items are arranged in the table with the three positively worded statements on the left and four negatively worded statements on the right.

**Positively-worded statements.** Providers obtained an overall mean score (M) of 4.4 for the item on being sensitive to the health care needs of migrants and refugees, indicating slightly more agreement than disagreement. Groups reporting lower scores for this item, included: the age category of 45–54 years (M 3.8; SD 1.8); enrolled nurses and nursing assistants (M 3.7; SD 1.6) and those working in a health care facility for a period of 5–9 years (M 3.8; SD 2.0). Conversely, providers born outside of South Africa (M 6.5; SD 0.8); medical doctors (M 5.5; SD 2.1) and allied health professionals (M 5.1; SD 3.1) had higher than average scores.

The highest level of agreement was obtained for the item on providing the same quality of care to migrants and refugees as to South Africans, with an overall mean score of 6.1 (SD 1.5). The lowest mean scores were reported from participants in the age categories 35–44 years (M 5.8; SD 1.6) and 45–54 years (M 5.9; SD 1.5); those who were single (M 5.8; SD 1.9); and

**Table 1. Demographic and employment characteristics of survey participants.**

| Variable | n | % |
|---|---:|---:|
| **Age** mean (SD) | 36.2 (11.4) | |
| **Age by category of health care professional** mean (SD) | | |
| Professional nurses | 45.0 (11.9) | |
| Enrolled nurses and nursing assistants | 38.3 (9.6) | |
| Medical doctors | 30.8 (7.1) | |
| Allied health professionals | 28.8 (6.9) | |
| All participants | 36.2 (11.4) | |
| **Age group (years)** | | |
| < 25 | 42 | 15.4 |
| 25–34 | 109 | 40.0 |
| 35–44 | 55 | 20.1 |
| 45–54 | 35 | 12.8 |
| 55+ | 32 | 11.7 |
| **Gender** | | |
| Female | 215 | 77.6 |
| Male | 62 | 22.4 |
| **Place of birth** | | |
| South Africa | 254 | 94.8 |
| Outside South Africa | 14 | 5.2 |
| **Marital status** | | |
| Single | 124 | 44.8 |
| Living together | 25 | 9.0 |
| Married | 108 | 40.0 |
| Divorced/ Widowed | 20 | 7.2 |
| **Category of health care professional** | | |
| Nurses: | | |
| Enrolled nurses | 30 | 10.8 |
| Nursing assistants | 33 | 11.9 |
| Professional nurses | 81 | 29.2 |
| All categories of nurses | 144 | 51.9 |
| Medical doctors | 70 | 25.3 |
| Allied health professionals: | | |
| Clinical associate | 1 | 0.4 |
| Social workers | 2 | 0.7 |
| Dieticians/ Dietician assistants | 9 | 3.2 |
| Pharmacists/ Pharmacist interns/ Pharmacist assistants | 24 | 8.7 |
| Radiographers | 6 | 2.2 |
| Rehabilitation therapists (audiologists, speech therapists) | 5 | 1.8 |
| All categories of allied health professionals | 47 | 17.0 |
| **Type of health care facility** | | |
| Central hospital | 72 | 25.1 |
| Clinic | 27 | 5.0 |
| Community health centre | 24 | 8.7 |
| District hospital | 36 | 13.0 |
| Regional hospital | 65 | 23.5 |
| Regional Tertiary hospital | 51 | 18.4 |
| Specialised Mother & Child hospital | 15 | 5.4 |

(*Continued*)

**Table 1.** (Continued)

| Variable | n | % |
|---|---|---|
| **Years worked in facility** mean (SD) | 6.8 (8.4) | |
| **Years worked in facility** median (range) | 3 (0.08–39) | |
| **Years worked in facility** | | |
| < 2 years | 82 | 29.6 |
| 2–4 years | 81 | 29.2 |
| 5–9 years | 57 | 20.6 |
| 10–14 years | 20 | 7.2 |
| 15+ years | 37 | 13.4 |

enrolled nurses/ nursing assistants (M 5.8; SD 1.3). Providers working in a health care facility for periods between 10 and 14 years (M 5.7; SD 1.1) and periods of more than 15 years (M 5.9; SD 1.5) also had lower mean scores.

Overall, the providers disagreed with migrants and refugees should be covered by the NHI, as indicated by a mean score of 3.4 (SD 2.0). Providers aged 45–54 years (M 2.8; SD 1.7), allied health professionals (M 2.7; SD 2.6), and those working in the facility for 10–14 years (M 2.8; SD 1.8) were more opposed to this proposal. The highest agreement score was obtained for providers born outside of South Africa (M 5.2; SD 2.1).

**Negatively-worded statements.** Overall, providers strongly disagreed that they discriminated against migrant and refugee patients, with a mean score of 1.7 (SD 1.1) (Table 2). Providers under 25 years disagreed most with this statement (M 1.1; SD 0.4), while those working in a health care facility for between 10–14 years disagreed least (M 2.3; SD 1.0), indicating relatively less and more exclusionary attitudes respectively.

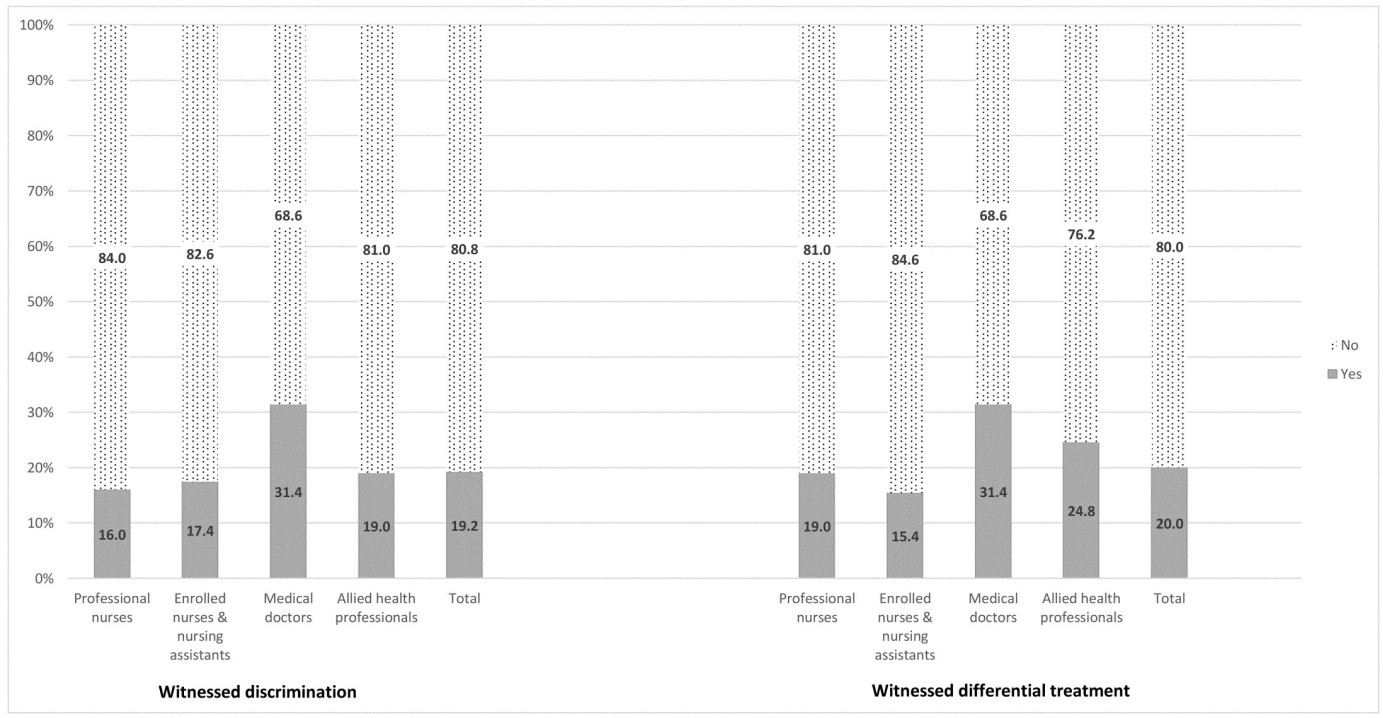

**Fig 1. Witnessed discrimination or differential treatment in workplace.**

 

**Table 2. Providers' mean scores (out of 7) of social exclusionary views or practices by socio-demographic and employment characteristics.**

| Variable | | I am sensitive to the health care needs of migrants and refugees | I provide the same quality of care to migrants and refugees as I do to South Africans | I believe migrants and refugees should be covered under the NHI | I discriminate against migrant and refugee patients | I have delayed health care to patients because of their migrant or refugee status | I believe migrant and refugee patients should go back to their home country for health care | I believe that migrant and refugee patients only come to South Africa for health care services |
|---|---|---|---|---|---|---|---|---|
| | | Mean (SD) | Mean (SD) | Mean (SD) | Mean (SD) | Mean (SD) | Mean (SD) | Mean (SD) |
| Total | | 4.4 (2.1) | 6.1 (1.5) | 3.4 (2.0) | 1.7 (1.1) | 1.7 (1.2) | 3.4 (2.1) | 3.6 (2.1) |
| Gender | Male | 4.5 (2.1) | 6.0 (1.8) | 4.0 (2.2) | 1.5 (1.0) | 1.5 (1.2) | 2.6 (1.9) | 3.1 (2.2) |
| | Female | 4.4 (2.1) | 6.1 (1.4) | 3.3 (2.0) | 1.7 (1.1) | 1.7 (1.2) | 3.6 (2.1) | 4.1 (2.1) |
| Age Group | < 25 years | 4.9 (2.7) | 6.7 (0.7) | 4.0 (2.7) | 1.1 (0.4) | 1.3 (1.0) | 2.7 (2.5) | 3.7 (2.3) |
| | 25–34 years | 4.5 (2.0) | 6.1 (1.6) | 3.7 (2.1) | 1.7 (1.3) | 1.6 (1.1) | 3.3 (2.1) | 3.5 (2.4) |
| | 35–44 years | 4.2 (2.1) | 5.8 (1.6) | 3.4 (2.0) | 1.7 (0.7) | 1.5 (0.6) | 3.2 (2.1) | 4.1 (2.0) |
| | 45–54 years | 3.8 (1.8) | 5.9 (1.5) | 2.8 (1.7) | 2.0 (1.0) | 2.0 (1.2) | 3.7 (2.0) | 4.5 (1.8) |
| | 55+ years | 4.7 (1.8) | 6.2 (1.1) | 3.4 (1.7) | 1.7 (0.8) | 2.2 (1.5) | 4.0 (1.7) | 4.0 (2.0) |
| Origin | Born in South Africa | 4.3 (2.0) | 6.1 (1.5) | 3.3 (2.0) | 1.7 (1.1) | 1.7 (1.2) | 3.5 (2.1) | 4.0 (2.2) |
| | Born outside South Africa | 6.5 (0.8) | 6.0 (2.2) | 5.2 (2.1) | 1.3 (0.5) | 1.2 (0.5) | 1.4 (1.6) | 2.6 (2.2) |
| Marital Status | Single | 4.2 (2.2) | 5.8 (1.9) | 3.2 (1.9) | 1.7 (1.0) | 1.7 (1.3) | 3.5 (2.2) | 4.2 (2.3) |
| | Living together | 4.5 (1.8) | 6.5 (0.9) | 4.0 (2.3) | 1.6 (1.1) | 1.5 (0.6) | 3.3 (2.0) | 3.3 (1.9) |
| | Married | 4.5 (2.2) | 6.3 (1.2) | 3.8 (2.2) | 1.6 (0.9) | 1.8 (1.3) | 3.2 (2.2) | 3.8 (2.2) |
| | Divorced/ Widowed | 4.2 (1.7) | 6.0 (1.2) | 3.4 (1.7) | 2.0 (1.0) | 1.8 (1.0) | 3.7 (1.8) | 2.6 (1.6) |
| HCP Category | Professional nurse | 4.5 (1.9) | 6.1 (1.4) | 4.3 (2.0) | 1.6 (0.8) | 1.7 (1.1) | 3.8 (2.0) | 4.3 (1.9) |
| | Enrolled nurse/ Nursing assistant | 3.7 (1.6) | 5.8 (1.3) | 4.0 (1.7) | 2.0 (1.0) | 2.0 (1.0) | 3.4 (1.6) | 3.9 (1.7) |
| | Medical doctor | 5.5 (2.1) | 6.4 (1.5) | 3.4 (2.4) | 1.3 (0.9) | 1.3 (0.9) | 2.7 (2.3) | 3.4 (2.4) |
| | Allied health professional | 5.1 (3.1) | 6.6 (1.4) | 2.7 (2.6) | 1.3 (1.0) | 1.3 (1.2) | 2.5 (2.8) | 2.6 (2.6) |
| Type of facility | Tertiary hospital | 4.2 (2.1) | 6.0 (1.6) | 4.1 (2.3) | 1.8 (1.1) | 2.0 (1.5) | 3.6 (2.2) | 3.2 (2.1) |
| | Regional hospital | 4.5 (2.2) | 6.0 (1.3) | 3.8 (2.1) | 1.8 (1.2) | 1.6 (0.8) | 3.0 (2.0) | 3.5 (2.0) |
| | District hospital | 4.2 (1.8) | 6.4 (0.5) | 3.4 (2.0) | 1.4 (0.6) | 1.5 (0.9) | 3.2 (1.9) | 3.9 (2.1) |
| | PHC facilities | 5.0 (2.1) | 6.0 (2.0) | 3.8 (2.1) | 1.3 (0.6) | 1.4 (1.0) | 3.6 (2.2) | 3.8 (1.8) |
| Years working at facility | < 2 years | 5.1 (2.3) | 6.0 (2.3) | 3.8 (2.6) | 1.4 (1.2) | 1.4 (1.1) | 3.1 (2.7) | 3.5 (2.5) |
| | 2–4 years | 4.6 (2.0) | 6.3 (1.1) | 3.5 (2.0) | 1.6 (1.2) | 1.5 (1.0) | 3.0 (2.0) | 3.5 (2.2) |
| | 5–9 years | 3.8 (2.0) | 6.1 (1.3) | 3.7 (2.0) | 1.7 (0.8) | 1.8 (1.1) | 3.4 (1.9) | 4.4 (2.0) |
| | 10–14 years | 4.4 (1.6) | 5.7 (1.1) | 2.8 (1.8) | 2.3 (1.0) | 2.0 (1.0) | 4.6 (1.9) | 4.5 (1.9) |
| | 15+ years | 4.4 (1.6) | 5.9 (1.5) | 3.0 (1.5) | 1.7 (0.9) | 2.0 (1.3) | 3.7 (1.8) | 4.0 (1.8) |

Providers also disagreed that they delayed health care to patients because of their migration status with an overall mean score of 1.7 (SD 1.2) (Table 2). The highest mean score for this item was 2.2 (SD 1.5) obtained from providers in the age category of 55 years and older, which

would still be interpreted as general disagreement with the statement (compared to the cutoff of 4), although they did disagree less than other groups.

Overall, providers agreed more that migrants and refugees should return to their home country for health care obtaining a mean score of 3.4 (SD 2.1) for that item. Lower mean scores were obtained for: male participants (M 2.6; SD 1.9); those younger than 25 years old (M 2.7; SD 2.5), those born outside of South Africa (M 1.4; SD 1.6); medical doctors (M 2.7; SD 2.3) and allied health professionals (M 2.5; SD 2.8).

Providers obtained an overall mean score of 3.60 (SD 2.1) for the item on migrants and refugees only coming to South Africa for health care services. Providers born outside South Africa (M 2.6; SD 2.2) and allied health professionals (M 2.6; SD 2.6) agreed least with this statement; while professional nurses actually agreed with it on average (M 4.3; SD 1.9).

## Predictors of social exclusionary views or practices among health care providers

Table 3 shows the results of the multiple regression analyses on the predictors of social exclusionary views or practices among health care providers. A negative co-efficient on the positively worded statements (first three) and a positive co-efficient on the negatively worded statements (last four) indicate relatively more exclusionary views (Table 3).

Participants born outside of South Africa had a significantly higher score (p<0.001) than those born in South Africa on being sensitive to the health care needs of migrant and refugee patients, indicative of less exclusionary views. Enrolled nurses and nursing assistants had a significantly lower score (p<0.001) than the reference category of allied health professionals on being sensitive to the health care needs of migrant and refugee patients, indicating more exclusionary views.

In relation to providing the same quality of care to migrant and refugee patients, participants aged 35–44 years (p = 0.04) held significantly more exclusionary views than the under 25 reference group, while enrolled nurses and nursing assistants (p = 0.01) were more exclusionary than allied professionals. With regard to the inclusion of migrants under the NHI, single participants had significantly lower mean scores (p = 0.01) than the married reference group, suggesting relatively more exclusionary views with regard to NHI coverage. Gender, age, category of health care professional, and years worked in health care facility were no longer significant contributors in the regression analysis for this item.

Providers aged 25–34 years (p = 0.01), 55 years and older (p = 0.004), enrolled nurses and nursing assistants (p = 0.01), and providers working in the health care facility for a period of between 10–14 years (p = 0.02) were significantly more likely than the reference groups to indicate agreement with having discriminated against migrant patients. In contrast, providers working in district hospitals and PHC facilities compared to tertiary hospitals agreed less with having discriminated against migrant patients (p = 0.02).

Category of health care professional and type of health care facility were predictors of participants' views on delaying care because of migration status. In particular, enrolled nurses and nursing assistants had a significantly higher score than allied health professionals indicating higher agreement that they had delayed care because of migration status (p = 0.03). On the other hand, participants from regional hospitals (p<0.001), district hospitals (p = 0.01) and PHC facilities (p<0.001) agreed less that they had delayed care than those from tertiary hospitals.

With regard to the view that migrants and refugees should return to their home country for health services, gender, category of health care professional, and place of birth were significant predictors. Female participants scored higher than men (p = 0.005), while professional nurses

**Table 3. Predictors of social exclusionary views or practices among health care providers.**

| Variable | | I am sensitive to the health care needs of migrants and refugees | | I provide the same quality of care to migrants and refugees as I do to South Africans | | I believe migrants and refugees should be covered under the NHI | | I discriminate against migrant and refugee patients | | I have delayed health care to patients because of their migrant or refugee status | | I believe migrant and refugee patients should go back to their home country for health care | | I believe that migrant and refugee patients only come to South Africa for health care or services | |
|---|---|---|---|---|---|---|---|---|---|---|---|---|---|---|---|
| | | β | p-value | β | p-value | β | p-value | β | p-value | β | p-value | β | p-value | β | p-value |
| **Gender** | Reference: Male | | | | | | | | | | | | | | |
| | Female | - | - | - | - | -0.65 | 0.18 | 0.15 | 0.37 | 0.09 | 0.73 | 0.95 | 0.005* | 0.87 | 0.03* |
| **Age Group** | Reference: < 25 years | | | | | | | | | | | | | | |
| | 25–34 years | 0.27 | 0.49 | -0.50 | 0.08 | -0.32 | 0.28 | 0.46 | 0.01* | 0.04 | 0.83 | 0.40 | 0.36 | -0.46 | 0.36 |
| | 35–44 years | 0.04 | 0.94 | -0.77 | 0.04* | -0.71 | 0.12 | 0.18 | 0.34 | -0.10 | 0.75 | -0.02 | 0.96 | -0.03 | 0.97 |
| | 45–54 years | 0.23 | 0.69 | -0.70 | 0.16 | -1.19 | 0.07 | 0.48 | 0.05 | 0.29 | 0.50 | 0.10 | 0.86 | 0.40 | 0.51 |
| | 55+ years | 1.36 | 0.16 | -0.35 | 0.38 | -0.45 | 0.51 | 0.59 | 0.004* | 0.78 | 0.11 | 0.55 | 0.50 | 0.02 | 0.98 |
| **Marital Status** | Reference: Married | | | | | | | | | | | | | | |
| | Single | - | - | 0.22 | 0.56 | -0.82 | 0.01* | - | - | 0.00 | 0.98 | - | - | 0.69 | 0.04* |
| | Living together | - | - | -0.60 | 0.17 | 0.01 | 0.98 | - | - | -0.13 | 0.65 | - | - | -0.32 | 0.45 |
| | Divorced/Widowed | - | - | -0.33 | 0.46 | -0.74 | 0.23 | - | - | -0.25 | 0.57 | - | - | -0.78 | 0.30 |
| **Born in/ out South Africa** | Reference: Born in SA | | | | | | | | | | | | | | |
| | Born outside SA | 1.58 | p<0.001* | - | - | - | - | 0.01 | 0.97 | -0.02 | 0.92 | -1.36 | p<0.001* | -1.17 | 0.02* |
| **HCP Category** | Reference: Allied health professional | | | | | | | | | | | | | | |
| | Professional nurse | -0.64 | 0.13 | -0.48 | 0.08 | -0.77 | 0.15 | 0.00 | 0.98 | 0.21 | 0.32 | 1.23 | 0.02* | 1.91 | p<0.001* |
| | Enrolled nurses & nursing assistants | -1.32 | p<0.001* | -0.73 | 0.01* | -0.64 | 0.18 | 0.54 | 0.01* | 0.49 | 0.03* | 1.13 | 0.009* | 1.42 | 0.03* |
| | Medical doctor | -0.13 | 0.67 | -0.19 | 0.44 | -0.56 | 0.23 | -0.08 | 0.71 | 0.06 | 0.78 | 0.69 | 0.22 | 1.65 | p<0.001 |
| **Type of facility** | Reference: Tertiary hospital | | | | | | | | | | | | | | |
| | Regional hospital | - | - | 0.10 | 0.72 | - | - | 0.01 | 0.98 | -0.40 | p<0.001* | - | - | - | - |
| | District hospital | - | - | 0.45 | 0.06 | - | - | -0.46 | 0.01* | -0.37 | 0.01* | - | - | - | - |
| | PHC facility | - | - | -0.00 | 0.99 | - | - | -0.37 | 0.04* | -0.52 | p<0.001* | - | - | - | - |
| **Years working at facility** | Reference: < 2 years | | | | | | | | | | | | | | |
| | 2–4 years | -0.06 | 0.81 | - | - | -0.21 | 0.61 | -0.08 | 0.66 | -0.07 | 0.73 | -0.55 | 0.25 | 0.01 | 0.98 |
| | 5–9 years | -0.46 | 0.27 | - | - | 0.41 | 0.48 | -0.10 | 0.58 | 0.13 | 0.74 | -0.26 | 0.65 | 0.63 | 0.27 |
| | 10–14 years | -0.01 | 0.98 | - | - | -0.31 | 0.64 | 0.58 | 0.02* | 0.31 | 0.40 | 0.72 | 0.20 | 0.45 | 0.52 |
| | 15 or more years | -1.16 | 0.21 | - | - | 0.01 | 0.98 | -0.23 | 0.59 | -0.18 | 0.74 | -0.41 | 0.34 | -0.26 | 0.38 |
| **Constant** | | 5.13 | p<0.001* | 7.26 | p<0.001* | 5.55 | p<0.001* | 1.13 | 0.001* | 1.43 | 0.018* | 1.73 | 0.001* | 1.54 | 0.022* |

Only predictor variables statistically significant at 20% in the bivariate analysis were included in the multiple linear regression models. *p<0.05

(p = 0.02), enrolled nurses and nursing assistants (p = 0.009) had significantly higher scores the allied professional reference group. Providers born outside of South Africa had significantly lower scores than those born in South Africa (p<0.001), indicating less exclusionary views.

Female providers had a significantly higher score than male providers on the view that migrants only come to South Africa for health care services (p = 0.03). Similarly, single participants also held more exclusionary views for this item (p = 0.04), compared to those who were

married or divorced. Professional nurses, (p<0.001), enrolled nurses and nursing assistants (p = 0.03), and medical doctors (p<0.001) had higher scores, indicating agreement with this view on migrants only coming to South Africa for health care.

## Discussion

This was the first survey in South Africa, and indeed in Africa, that we know of that quantitatively examined the perspectives of health care providers on public health care services to migrants. Most of the study participants were female (77.6%) and nurses (51.9%). This is not surprising as the majority of health care providers in South Africa are nurses, and women [32]. In 2018, the South African Nursing Council (SANC) statistics showed that 90% of nurses were women [33]. Our study showed that the variables that influenced health care providers' views on social exclusion were gender, marital status, age, category of health care provider, type of facility and years worked in a health care facility (Table 3).

Almost one in five health care providers (19.2%) reported that they had witnessed discrimination, and 20.0% reported that they had witnessed differential treatment of migrants in their work settings (Fig 1). Medical doctors were more likely to report that they had witnessed discrimination of migrant patients in their workplace. A possible explanation for this finding is that medical doctors may be more aware of acts of discrimination against migrants and thus, able to notice such practices when it occurs. Importantly, these findings suggest that medical xenophobia occurs, as described in a previous study in the South African public health system [34]. Studies in other countries have shown variable findings on health care providers' perspectives on discrimination. For example, European studies that examined health care providers' experiences of discrimination found both a reluctance to talk about discrimination, and evidence of discriminatory attitudes towards migrant patients [35, 36]. However, another study in Greece found a more mixed picture of the interactions between providers and migrants, with some providers prioritizing the health care of citizens over migrants, while others provided unrestricted health care access to undocumented migrants despite restrictive laws [37].

In our study, health care providers indicated some sensitivity to the health needs of migrants (mean score 4.4). Providers born outside South Africa expressed greater sensitivity to the needs of migrants. This is not surprising, because these providers are also migrants, and hence identified with migrant patients. In a very different context, a Canadian study found that providers born outside Canada expressed greater cultural sensitivity and were more comfortable with immigrant patients than Canadian-born health providers [38].

Enrolled nurses and nursing assistants in our study had lower scores for sensitivity to the needs of migrants, suggesting that their views were more exclusionary. In South Africa, enrolled nurses undergo two years of training and nursing assistants undergo one year of training. The relatively short training period might be insufficient to instill ethics, values, and culturally responsive health care, which would partly explain these findings. Further research is needed to explore whether the content and length of pre-service training influences views on migrant patients. Studies in Canada and Australia, albeit with physicians, found that cultural barriers hindered the provision of migrant-sensitive health care services [39–41]. Several studies in the same countries have demonstrated that culture-sensitivity training of health care providers can improve the health outcomes of migrant patients [42], through better expressed sensitivity [43, 44], empathy [45] and cultural humility [46]. Although the context of these studies is different from that of South Africa, there would be value in ethics and culture-sensitivity training for all health care providers in the South African public service, with a particular focus on enrolled nurses and nursing assistants.

The pattern of health provider responses was complex and nuanced. For example, although they reported ambivalence on sensitivity to the health care needs of migrants (mean score 4.4), they strongly agreed that they provided the same quality of care to migrants as to South Africans (mean of 6.1). This divergence might suggest that they honor their professional and ethical obligations, even though they may not be that sympathetic to the needs of migrants. More research is needed to explore these apparent differences and contradictions.

Health care providers in our study strongly disagreed that they discriminated against migrants, or delayed care to migrants. Both of these findings are encouraging. However, there were significantly higher mean scores for these two items among enrolled nurses and nursing assistants, indicating more discriminatory attitudes and practices in this group. Other factors associated with discriminating against migrants included providers aged 25–34, older than 55, and working in a health care facility between 10 and 14 years. Conversely, we also found that working in a district hospital and in a PHC facility were predictors of less exclusionary views.

The age variation in discrimination against migrants requires further research. I Health providers in the oldest age category (>55 years) reported more discriminatory views. Another study found that older individuals tend to hold more conservative views [47]. In South Africa, the widespread promotion of human rights has occurred in recent history since the country's democracy, and may have less traction in older generations. The discriminatory attitudes in the younger age groups is more concerning. The findings suggest that legislative changes such as South Africa's Bill of Rights are essential, but do not guarantee changes in attitudes or behaviours. The effect of age on discriminatory views on migrants has also been found in a 2019 survey on social cohesion by the Gauteng City-Region Observatory (GCRO) in South Africa [48]. The study found that participants 55 years and older were more inclined to agree with the view that migrants should be sent home, while respondents between the ages of 25 and 39 years old were more likely to accept/endorse violence against foreigners [48].

The expressed discriminatory views in our study are concerning, as health care providers are required to uphold professional and ethical standards of care [49, 50]. The various health professional Oaths emphasis service to humanity, practicing with conscience, treating all patients with dignity, pursuing justice, and advocating on behalf of vulnerable and disadvantaged patients [51, 52]. A combination of strategies is needed to ensure that migrant-sensitive health services are provided, and that all patients in the Gauteng public health service are treated with respect and dignity, regardless of nationality. These strategies include migrant-inclusive health policies specifically on the entitlements of migrants and health care, advocacy training and campaigns that emphasise the rights and responsibilities of providers, lobbying by civil society organisations, and clear communication about the complaints mechanisms, including the Office of the Health Ombud [53]. There should also be adverse consequences for those health care providers that continue discrimination against migrants, and they should be reported to the relevant health professions council for possible disciplinary action.

The mean score for the item that migrants should return to their home country for health care was 3.4, and that migrants only come to South Africa for health care was 3.6. Female providers, professional nurses and enrolled nurses and nursing assistants had significantly higher agreement than the other categories, again suggesting more social exclusionary perspectives. These views could explain our finding that participants did not agree with the inclusion of migrants and refugees in the proposed NHI scheme (mean score of 3.4). In the regression, single status was the only significant predictor of a more exclusionary view, but it is unclear why this was the case. It is still of concern that most health care providers hold this view, given that they have a critical role to play in the achievement of UHC [54]. Scholars have suggested that health care providers work in constrained conditions, exacerbated by migrant-unfriendly regulatory frameworks, policies and the political rhetoric of government officials that amplify

xenophobic sentiments in South African society [55, 56]. Health care providers may reflect the political rhetoric of xenophobia [57, 58]. This context might explain the social exclusionary views of some of the health workers in our survey. Moreover, agreeing that migrants only come to South Africa for health care may not indicate anti-migrant attitudes, but may reflect the current reality given the virtual collapse of the health systems in their home countries, as in the case in Zimbabwe [59].

The study is cross-sectional and reflects health care providers' perspectives at a point in time. Future research could explore whether and how health care providers' perspectives change over time. The self-reported information obtained from health care providers may be influenced by social desirability bias. However, the self-administered questionnaire using tablets allowed providers to express their views in a confidential manner. The study was only conducted in Gauteng Province, and the findings might not apply to other provinces.

However, there are numerous study strengths. Firstly, this was one of the first surveys that examined the perspectives of health care providers on migrants utilising public health facilities. Secondly, the study quantified social exclusionary attitudes or behaviours and reported discrimination by health care providers against migrant patients, as well as the factors that influence discrimination (e.g. category of health worker). Lastly, the findings provide a baseline for future studies, both quantitative and qualitative, in South Africa as well as in similar low- and middle-income countries. Future research could include a larger sample of health care providers, as well as qualitative research that could provide more depth to the views of different categories of health care providers.

As South Africa moves towards the implementation of the NHI, discrimination and other social exclusionary views or practices of health care providers will undermine progress. The United Nations has called on Member States to put an end to discrimination of migrants in health care settings. There are three key priority areas of action: supporting the rights of both patients and providers; tackling discrimination through evidence and appropriate legal frameworks that ensure accountability; and lastly, collaboration between governments, civil society and communities to address the determinants of discrimination [60]. Health care providers are the foundation of quality UHC. Our study findings indicate that the interaction between providers and migrant patients in Gauteng are complex, and there is no straightforward or single narrative. Given its economic importance, Gauteng should take the lead in implementing the UN recommendations, and should develop more inclusive health policies that are more in consort with the Constitution of South Africa to the benefit of all patients at all levels of care, in support of the achievement of UHC.

## Conclusion

Health care providers, specifically medical doctors and nurses, are at the front-line of health care delivery and are thus integral to the provision of migrant-sensitive health care, and the achievement of UHC. Given this criticality, providers' perspectives on social exclusionary views or practices are important in shaping inclusive health policies. Using a lens of social exclusion, we have generated new knowledge on health care provider-migrant interactions in the Gauteng province of South Africa. Social exclusionary views or practices must be addressed at all levels through inclusive health policies, training in culture-sensitivity, ethics and human rights; and promoting health care providers as advocates for migrant patients and their rights.

## Supporting information

**S1 Questionnaire.**
(DOCX)

## Acknowledgments

We thank Dr Bridget Ikalafeng for facilitating permission at Provincial level to conduct the study. We also thank the chief executive officers and facility managers for permission to conduct the study at facilities and the ease of access their support provided. We acknowledge with gratitude the statistical advice and inputs of Professor Jonathan Levin and Dr Innocent Maposa. We thank the five fieldworkers who assisted with data collection.

## Author Contributions

**Conceptualization:** Janine A. White, Laetitia C. Rispel.

**Data curation:** Janine A. White.

**Formal analysis:** Janine A. White, Duane Blaauw, Laetitia C. Rispel.

**Investigation:** Janine A. White.

**Methodology:** Janine A. White, Laetitia C. Rispel.

**Project administration:** Janine A. White.

**Supervision:** Laetitia C. Rispel.

**Validation:** Janine A. White.

**Visualization:** Janine A. White.

**Writing – original draft:** Janine A. White.

**Writing – review & editing:** Janine A. White, Duane Blaauw, Laetitia C. Rispel.

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
