## [Decision Letter · Decision Letter 0]

18 May 2020

PONE-D-19-34029

Social exclusion and perceptions of health care providers on migrants in Gauteng public health facilities, South Africa

PLOS ONE

Dear Ms White,

Thank you for submitting your manuscript to PLOS ONE. After careful consideration, we feel that it has merit but does not fully meet PLOS ONE’s publication criteria as it currently stands. Therefore, we invite you to submit a revised version of the manuscript that addresses the points raised during the review process.

This is clearly a timely topic and both reviewers see merit in this study. We ask that you clearly address each of the points made by the reviewers and give the additional details as requested. PLOS one is an international journal and so please ensure that the context of migration in South Africa is addressed, as this requires further explanation and has been highlighted by both of the reviewers. 

We would appreciate receiving your revised manuscript by Jul 02 2020 11:59PM. To enhance the reproducibility of your results, we recommend that if applicable you deposit your laboratory protocols in protocols.io, where a protocol can be assigned its own identifier (DOI) such that it can be cited independently in the future. For instructions see: http://journals.plos.org/plosone/s/submission-guidelines#loc-laboratory-protocols

We look forward to receiving your revised manuscript.

Kind regards,

Fiona Cuthill, PhD

Academic Editor

PLOS ONE

Journal Requirements:

2. Please include additional information regarding the survey or questionnaire used in the study and ensure that you have provided sufficient details that others could replicate the analyses.

For instance, if you developed a questionnaire as part of this study and it is not under a copyright more restrictive than CC-BY, please include a copy, in both the original language and English, as Supporting Information.

Moreover, please include more details on how the questionnaire was pre-tested, and whether it was validated.

3. Please provide additional details regarding participant consent.

In the ethics statement in the Methods and online submission information, please ensure that you have specified (i) whether consent was informed and (ii) what type you obtained (for instance, written or verbal).

If your study included minors, state whether you obtained consent from parents or guardians.

If the need for consent was waived by the ethics committee, please include this information.

4. In your Methods section, please provide additional information about the participant recruitment method and the demographic details of your participants. Please ensure you have provided sufficient details to replicate the analyses such as: a) the recruitment date range (month and year), b) a description of any inclusion/exclusion criteria that were applied to participant recruitment, c) a table of relevant demographic details, and d) a statement as to whether your sample can be considered representative of a larger population.

5. Our internal editors have looked over your manuscript and determined that it is within the scope of our Health Inequities and Disparities Research Call for Papers.

This collection of papers is headed by a team of Guest Editors for PLOS ONE: Clare Bambra, Hans Bosma, Diana Burgess, Joseph Telfair, Barbara Turner, and Jennie Popay. The Collection will encompass a diverse range of research articles on health inequities and disparities. 

Additional information can be found on our announcement page: hhttps://collections.plos.org/s/health-inequities

If you would like your manuscript to be considered for this collection, please let us know in your cover letter and we will ensure that your paper is treated as if you were responding to this call.

If you would prefer to remove your manuscript from collection consideration, please specify this in the cover letter.

6. We note that you have indicated that data from this study are available upon request. PLOS only allows data to be available upon request if there are legal or ethical restrictions on sharing data publicly. For information on unacceptable data access restrictions, please see http://journals.plos.org/plosone/s/data-availability#loc-unacceptable-data-access-restrictions.

7. Please include a copy of Table 4 which you refer to in your text on page 20.

<h3>** **</h3>

8. We note you have included a table to which you do not refer in the text of your manuscript. Please ensure that you refer to Table 3 in your text; if accepted, production will need this reference to link the reader to the Table.

Reviewers' comments:

Reviewer's Responses to Questions

**Comments to the Author**

1. Is the manuscript technically sound, and do the data support the conclusions?

Reviewer #1: Yes

Reviewer #2: Partly

2. Has the statistical analysis been performed appropriately and rigorously? 

Reviewer #1: Yes

Reviewer #2: Yes

3. Have the authors made all data underlying the findings in their manuscript fully available?

Reviewer #1: Yes

Reviewer #2: Yes

4. Is the manuscript presented in an intelligible fashion and written in standard English?

Reviewer #1: Yes

Reviewer #2: Yes

5. Review Comments to the Author

Reviewer #1: This is well researched and compelling paper that addresses an important topic. I recommend publication with minor revisions.

There are two areas that need further development/explanation:

First, it would be helpful to further explain the context of data collection and the methods of access and introduction by which the field workers were engaged with the health care providers. Under “Measures,” were the field workers standing next to the health care providers answering questions as they were filling out the questions on the tablet? What were the ethnicities of the field workers? Could cultural differences have played into how the health care providers responded? Were the health care providers offered anything in return for participating in the study? Were supervisors present as providers filled out the questionnaire? It is assumed that the health care providers were taking time from work at their workplace to participate. Such questions should be addressed because the contextual information may pertain to some variations in the responses and efficacy of the study.

Second, although the Social Exclusion Knowledge Network (SEKN) is an interesting framework, it excludes the workplace dynamics that follow from the laws that apply to health care providers, and the extension of care to foreigners. Specifically, South African law, under the National Health Act of 1998 explicitly mentions progressive realisation – the notion that the State pays for what it can. This element is critical in providing care of foreigners in South Africa. It is not clear from the conclusions whether the authors are advocating that hospital provide care for all foreigners or rather just provide cultural sensitivity training.

Perhaps most importantly, the paper neglects to mention how South Africans who assist illegal foreigners are held liable. Some hospital administrators have been known to pressure health care providers with this legal provision to refuse care. Therefore, where the questions were asked and the context are vital to better understand the data collected. The authors would be served by better understanding the complexity of South African laws and the divergence between the Constitution and immigration law with respect to health care access. (This could be discussed in line 77.)

Immigration Statute under Section 49 that says that anyone who intentionally facilitates an illegal foreigner in public services is held liable. Although the immigration statute indicates under Section 49 (4) that, “anyone who intentionally facilitates an illegal foreigner to receive public services to which such illegal foreigner is not entitled shall be guilty of an offence and liable on conviction to a fine.” (Immigration Act 13 of 2002, Section 44 substituted by Section 42 of Act 19 of 2004, p. 52).

This article may help the authors in providing some additional context. https://www.emerald.com/insight/content/doi/10.1108/IJMHSC-04-2015-0012/full/html

The statistical evidence of the paper supports intervention to remediate discriminatory and social exclusionary views among health care providers

Reviewer #2: This paper is a quantitative study on the perspectives of Gauteng healthcare workers of migrants and ‘social exclusion’; a term which remains somewhat vague, i.e. a theoretical framework of ‘social exclusion’ is mentioned early on, but is never really applied in any analytical or conceptual sense. The paper does not detail who ‘migrants’ are; their numbers; where they come from; or the multiple challenges they face in South African society. This provides an overview: Steenberg 2020; http://medanthrotheory.org/read/11896/structural-vulnerabilities-and-healthcare-services-integration

The paper essentially informs that “21.0% of health care providers reported that they had witnessed discrimination against migrants, while 22.6% reported differential treatment of migrant patients. However, it is not at all specified what ‘discrimination’ or ‘differential treatment’ actually refers to; what it consists of; how and where it is enacted; degrees of severity; and/or consequences for patient outcomes, et cetera. Apparently, the study did include qualitative, open-ended questions “to allow for any additional comments […], but the qualitative information is excluded from this paper” --- which seems like a missed opportunity.

It is further reported that “Enrolled nurses and nursing assistants were significantly more likely to agree with social exclusionary views or practices”, but again there’s no real evidence-based explanation given as why that is or what it means (?).

In short, I think this is a lot of work (and words) to say disappointingly little --- be it about (non-descript) migrants; the (undefined) discrimination they suffer; or contrasts in the perceptions of healthcare workers.

This is a fascinating and timely issue and adding more detail, data, and discussion about migrants and their plights in the healthcare sector could make this a richer read, especially if (the omitted) qualitative data could inform how migrants are differentially treated and discriminated against --- and perhaps what this means for their engagement with SA healthcare services --- or what moves and motivates healthcare workers’ behaviors.

I’ve made a few comment in the PDF margins, but this is my main assessment. I would either reject or ask for revisions. Best of luck with the paper.

6. PLOS authors have the option to publish the peer review history of their article (what does this mean?). If published, this will include your full peer review and any attached files.

Reviewer #1: Yes: Theresa Alfaro-Velcamp, PhD

Reviewer #2: No

---

## [Author Response · Author response to Decision Letter 0]

7 Jul 2020

We would like to thank the reviewer and editor for their comments. We have addressed the reviewers and editor comments in extensive detail in the document labelled Cover Letter/ Response to Reviewers, uploaded in the attached files section. Our responses are outlined in table format with corresponding page and line number of changes (where applicable) and would be best viewed in the document uploaded in the attached files section.

---

## [Decision Letter · Decision Letter 1]

7 Oct 2020

PONE-D-19-34029R1

Social exclusion and the perspectives of health care providers on migrants in Gauteng public health facilities, South Africa

PLOS ONE

Dear Dr. White,

Thank you for submitting your manuscript to PLOS ONE. After careful consideration, we feel that it has merit but does not fully meet PLOS ONE’s publication criteria as it currently stands. Therefore, we invite you to submit a revised version of the manuscript that addresses the points raised during the review process.

Apologies for the delay in securing a third reviewer for your paper but this has now been completed. The reviewer has raised some issues in relation to updating the literature review and in clarifying whether your research on health inequalities is a comparison with European or non-European countries. Please address all of these comments in full and I look forward to your final submission. 

As the manuscript was submitted to the  Call for Papers on Health Disparities and Inequities, Guest Editor Diana Burgess also assessed it and considered it for inclusion in the Collection. Please find her comments below, and consider them in your revision:<o:p></o:p>

"I think it manuscript is suitable for inclusion. I also think the author was very responsive to the reviewer concerns. I had some minor concerns with the discussion section but these do not diminish my enthusiasm for the manuscript. The discussion presents specific results (e.g., "Health care providers obtained a score of 3.4 for the item on sensitivity to the health needs of migrants; and a score of 6.1 for providing the same quality of care"), rather than providing a broader interpretation of their findings. I also would like the authors to provide potential explanations for the effects of provider age on their attitudes and beliefs about migrants, since the effects of age were not clear cut and not consistent across items. Overall, I think this manuscripts makes a nice contribution to the literature and fills a gap in research about healthcare for migrant individuals.”<o:p></o:p>

We look forward to receiving your revised manuscript.

Kind regards,

Fiona Cuthill, PhD

Academic Editor

PLOS ONE

Reviewers' comments:

Reviewer's Responses to Questions

**Comments to the Author**

1. If the authors have adequately addressed your comments raised in a previous round of review and you feel that this manuscript is now acceptable for publication, you may indicate that here to bypass the “Comments to the Author” section, enter your conflict of interest statement in the “Confidential to Editor” section, and submit your "Accept" recommendation.

Reviewer #3: All comments have been addressed

2. Is the manuscript technically sound, and do the data support the conclusions?

Reviewer #3: Yes

3. Has the statistical analysis been performed appropriately and rigorously? 

Reviewer #3: Yes

4. Have the authors made all data underlying the findings in their manuscript fully available?

Reviewer #3: Yes

5. Is the manuscript presented in an intelligible fashion and written in standard English?

Reviewer #3: No

6. Review Comments to the Author

Reviewer #3: The paper provides significant evidence about the potentially discriminatory views of health-care providers in South Africa. However, although the authors have addressed the comments of reviewers to a large extent, the revised manuscript fails to address the concept of social exclusion in health, in a coherent way. Health care provision varies across countries and time periods. It is unclear whether the paper intends to make a comparison of the South-African case with other European or non-European countries. Therefore, the examples of Canada and Greece are not very helpful in documenting the disparities in health care for migrant populations.

The literature on social exclusion has to be updated. The authors may have a look at a newer report by WHO (World Health Organization Regional Office for Europe. Poverty and Social Exclusion in the WHO European Region: Briefing on policy issues produced through the WHO/European Commission equity project: World Health Organization Regional Office for Europe. 2010) and relevant bibliography on social exclusion (i.e O’Donnell, P., O’Donovan, D. & Elmusharaf, K. Measuring social exclusion in healthcare settings: a scoping review. Int J Equity Health 17, 15 (2018), or the work of MIPEX health strand consortium , Ingleby et. Al. European Journal of Public Health, Volume 29, Issue 3, June 2019, Pages 458–462.

There are also more recent studies addressing the health inequalities in Greece (e.g Health inequalities among migrant and native-born populations in Greece in times of crisis: the MIGHEAL study European Journal of Public Health, Volume 28, Issue suppl_5, December 2018, Pages 5–19, https://doi.org/10.1093/eurpub/cky225)

Minor corrections:

in p. 3, l. 7 it is unclear where the “conservative” refers to. In addition, in p. 5 lines 107-109 where “all these challenges” refer to. The manuscript needs language editing.

7. PLOS authors have the option to publish the peer review history of their article (what does this mean?). If published, this will include your full peer review and any attached files.

Reviewer #3: No

---

## [Author Response · Author response to Decision Letter 1]

18 Nov 2020

We have addressed the reviewers' comments in our cover letter.

---

## [Editor Report · Decision Letter 2]

3 Dec 2020

Social exclusion and the perspectives of health care providers on migrants in Gauteng public health facilities, South Africa

PONE-D-19-34029R2

Dear Dr. White,

We’re pleased to inform you that your manuscript has been judged scientifically suitable for publication and will be formally accepted for publication once it meets all outstanding technical requirements.

Kind regards,

Fiona Cuthill, PhD

Academic Editor

PLOS ONE
---

## [Editor Report · Acceptance letter]

7 Dec 2020

PONE-D-19-34029R2 

Social exclusion and the perspectives of health care providers on migrants in Gauteng public health facilities, South Africa 

Dear Dr. White:

I'm pleased to inform you that your manuscript has been deemed suitable for publication in PLOS ONE. Congratulations! Your manuscript is now with our production department. 

Kind regards, 

on behalf of

Dr. Fiona Cuthill 

Academic Editor

PLOS ONE